# Molecular Cluster Mining of Adrenocortical Carcinoma via Multi-Omics Data Analysis Aids Precise Clinical Therapy

**DOI:** 10.3390/cells11233784

**Published:** 2022-11-26

**Authors:** Yu Guan, Shaoyu Yue, Yiding Chen, Yuetian Pan, Lingxuan An, Hexi Du, Chaozhao Liang

**Affiliations:** 1Department of Urology, The First Affifiliated Hospital of Anhui Medical University, 218th Jixi Road, Hefei 230022, China; 2Institute of Urology, Anhui Medical University, 81th Meishan Road, Hefei 230022, China; 3Anhui Province Key Laboratory of Genitourinary Diseases, Anhui Medical University (AHMU), 81th Meishan Road, Hefei 230022, China; 4Department of General, Visceral, and Transplant Surgery, Ludwig-Maximilians-University Munich, D-81377 Munich, Germany

**Keywords:** adrenocortical carcinoma, multi-omics analysis, prognosis and treatment, cell signaling pathway, sensitivity to drugs

## Abstract

Adrenocortical carcinoma (ACC) is a malignancy of the endocrine system. We collected clinical and pathological features, genomic mutations, DNA methylation profiles, and mRNA, lncRNA, microRNA, and somatic mutations in ACC patients from the TCGA, GSE19750, GSE33371, and GSE49278 cohorts. Based on the MOVICS algorithm, the patients were divided into ACC1-3 subtypes by comprehensive multi-omics data analysis. We found that immune-related pathways were more activated, and drug metabolism pathways were enriched in ACC1 subtype patients. Furthermore, ACC1 patients were sensitive to PD-1 immunotherapy and had the lowest sensitivity to chemotherapeutic drugs. Patients with the ACC2 subtype had the worst survival prognosis and the highest tumor-mutation rate. Meanwhile, cell-cycle-related pathways, amino-acid-synthesis pathways, and immunosuppressive cells were enriched in ACC2 patients. Steroid and cholesterol biosynthetic pathways were enriched in patients with the ACC3 subtype. DNA-repair-related pathways were enriched in subtypes ACC2 and ACC3. The sensitivity of the ACC2 subtype to cisplatin, doxorubicin, gemcitabine, and etoposide was better than that of the other two subtypes. For 5-fluorouracil, there was no significant difference in sensitivity to paclitaxel between the three groups. A comprehensive analysis of multi-omics data will provide new clues for the prognosis and treatment of patients with ACC.

## 1. Introduction

Adrenocortical carcinoma (ACC) is an aggressive endocrine malignancy that originates in the adrenal cortex, accounting for approximately 5% of adrenal tumors, with an annual incidence of 0.7–2.0 cases per million people [1,2]. The onset age of ACC has bimodal characteristics, with a high incidence in the two age stages of 40–50 years old and 1–5 years old [2,3]. The disease stage is one of the most important prognostic factors. Currently, the staging system proposed by the European Adrenal Neoplasms Research Network (ENSAT) is a commonly used international standard [4]. Stages I and II are confined to the organs and can be cured by complete resection. Stages III and IV are considered aggressive and metastatic advanced tumors, and the five-year survival rate for stage IV patients is only 6–13% [2,5]. Some researchers believe that ACC may be related to the overproduction of steroid precursors, which have the characteristics of early metastasis and recurrence [6]. Complete surgical removal of ACC is the only chance for a long-term cure [7]. Mitotane has cytotoxic effects on steroidogenic cells in the adrenal cortex; therefore, it is recommended as a clinical adjuvant therapy, but its therapeutic effect is still unsatisfactory [2,8].

Some researchers have found that insulin-like growth factor 2 (IGF2), β-catenin (CTNNB1), and TP53 may be potential drivers of sporadic adrenocortical tumors. The IGF system has growth-promoting and differentiation functions in the adrenal glands. IGF2 overexpression is found in most ACCs and is often associated with poor outcomes [1,9]. It regulates cell proliferation and apoptosis mainly by binding to the insulin-like growth factor 1 receptor (IGF1R), especially in pediatric patients [10,11]. As a key component of the Wnt signaling pathway, β-catenin plays an important role in the development of the adrenal cortex [12], and is a poor prognostic factor in ACC [13]. Somatic mutations are common in ACC, and some researchers have found that somatic mutations in CTNNB1 are independent predictors of poor disease-free survival and overall survival in ACC [14]. Pan-genomic studies have found that TP53 mutations, mainly exon mutations, are common in sporadic ACC cases [15].

Multiple high-throughput detection techniques have been used in multi-omics association studies to elaborate on a single scientific subject. Numerous variables can affect cancer development. These include a wide range of information on various facts. Information may be combined at several levels and integrated, and staging prediction accuracy can be improved by multi-omics association studies [16]. Over the past 50 years, the genetic approach to cancer has taken over the profession. However, this genome-only perspective is limited and has the propensity to present cancer as a strongly heritable illness. According to new research, cancer is a multi-omics illness and is not as heritable or exclusively hereditary as previously believed. Cancer development and manifestation are influenced by the exposome, metabolome, and genome. Cancer-specific metabolism has been genetically altered to feed and support proliferating cancer cells [17]. The etiology of tumor growth may only be partially understood using a single type of molecular dataset. Several studies have used multi-omics data to categorize cancer patients and predict prognoses [18,19]. Therefore, a crucial stage in the machine-learning-model-based prediction of survival and recurrence is learning new characteristics from multi-omics data that help predict prognosis.

Multi-omics analysis of ACC can reveal several undiscovered oncogenic alterations and guide the exploration of new therapeutic approaches. In this study, we collected clinical and pathological characteristics, DNA methylation profiles, genomic mutations, and mRNA, lncRNA, and microRNA (miRNAs) information of patients, and somatic mutation data from four ACC datasets. Multi-omics analysis using the MOVICS algorithm provided new clues for the prognosis and treatment of patients with ACC.

## 2. Materials and Methods

### 2.1. Data Collection

Multi-omics data of ACC patients, including DNA methylation, gene mutations, mRNA, miRNA, and lncRNA, were downloaded from the TCGA-ACC dataset for molecular subtyping. The “TCGAbiolinks” R package (version 2.25.3) which is provided in https://github.com/BioinformaticsFMRP/TCGAbiolinks.git, was used to obtain clinical features and transcriptomic expression data. Gene symbol annotation was performed as described in our previous study [20]. The miRNA expression and DNA methylation 450 matrices were downloaded from the UCSC Xena (https://xenabrowser.net/datapages, accessed on 15 April 2022). Somatic mutation data were downloaded from the cbiopportal (https://www.cbioportal.org/, accessed on 15 April 2022). After combining all available data from the different patients, 78 eligible patients were included in the TCGA-ACC cohort. In addition, we extracted data from 89 patients with ACC from the GSE19750, GSE33371, and GSE49278 cohorts [21,22,23,24]. The batch effect is an abiotic difference between at least two datasets. To eliminate the bias caused by the batch effect, we used the combat algorithm package “SVA” (version 3.46.0). The GEO combined cohort was used as the subsequent test cohort, and the TCGA-ACC cohort was used as the training cohort.

### 2.2. Molecular Subtypes were Identified Using Multi-Omics Analysis

Molecular subtypes were determined using multi-omics data according to recently published guidelines for the R package “MOVICS” (version 1.0) [25]. The input for “MOVICS” is multi-omics data, and the output is the recommended molecular subtypes, presenting molecular features, prognosis, treatment sensitivity, and others; the VIGNETTE of this package is provided in https://xlucpu.github.io/MOVICS/MOVICS-VIGNETTE.html (accessed on 15 April 2022), with details of how to use it. First, univariate Cox regression analysis was used to evaluate factors related to overall survival (OS), including relevant biological information downloaded as above (all *p* < 0.05). Mutated genes were those with a mutation frequency greater than 10%. Based on the evaluation of the multi-omics data, the cluster prediction index (CPI) [26] and the gap statistic [27] were utilized to determine the correct number of subtypes. The number that resulted in the maximum value of both the gap statistic and the CPI was chosen as the optimum number of clusters for the input data. Ten clustering algorithms (iClusterBayes, moCluster, CIMLR, IntNMF, ConsensusClustering, COCA, NEMO, PINSPlus, SNF, and LRA) were then used to separate patients into various subtypes, and a combined classification using a consensus set was used to identify each subtype with a high degree of robustness. 

Specifically, if $t_{max}$ algorithms are specified where 2 ≤ $\;t_{max}$ ≤ 10, the package calculates a matrix $M\frac{\left(t\right)}{n\times n}$ per algorithm, where *n* is the number of samples and $M\frac{\left(t\right)}{ij}=1$ when samples *i* and *j* are clustered in the same subtype; otherwise, $M\frac{\left(t\right)}{ij}=0$. After obtaining all results from specified algorithms, MOVICS calculates a consensus matrix $CM=\sum_{t=1}^{t_{max}}M^{\left(t\right)}$, and $CM_{ij}$∈ [0, 10]. The sample similarity among the subtypes was calculated using silhouette scores.

### 2.3. Characteristics of Genetic Variations among Subtypes

Tumor-mutation burden (TMB) is the number of mutations per million bases, and fraction genome alteration (FGA) refers to the percentage of gene fragments with increased or lost copy numbers in the total genome. Total neoantigen and cytolytic activity (CYT) scores from previous studies were predicted by analyzing tumor-specific mutations, splicing, gene fusions, endogenous reverse transcription factors, and other criteria [28]. The downloaded copy number data from FireBrowse (http://firebrowse.org/, accessed on 15 April 2022) were visualized using the MafTools R package (version 2.14.0).

### 2.4. Comparison of Signaling Pathway Activation and Immune Infiltration

Single-sample gene-set enrichment analysis (ssGSEA) [28] package “GSVA” (version 1.1.11) was used to analyze 50 HALLMARK gene sets for each patient to reveal the activation of biological pathways. The Molecular Signatures Database (MSigDB, accessed on 15 April 2022) is one of the most widely used and comprehensive databases of gene sets for performing gene set enrichment analyses. The developers used a combination of automated approaches and expert curation to develop a collection of “hallmark” gene sets as part of the MSigDB. Each hallmark in this collection consists of a “refined” gene set derived from multiple “founder” sets that convey a specific biological state or process and display coherent expression [29]. The enrichment score (ES) represents the main result of gene enrichment analysis. In the ranking list, the ES-positive gene sets were at the top, and the ES-negative gene sets were at the bottom. The normalized enrichment score (NES) was the main evaluation index of the gene set enrichment results. The false discovery rate (FDR) is the rate of errors occurring in all discoveries with a set threshold of 0.05. We evaluated the immune infiltration of immunocytes and the activated status of the immune signature. Gene sets were collected from prior studies. The NES scores of different subgroups were calculated from the gene sets associated with immune and stromal features extracted from previous studies to demonstrate differences in the immune activation status [30]. We also used ssGSEA to study the infiltration of 28 immune cells in tumors and calculated the infiltration score of each immune cell in each patient [31]. Metabolism-associated pathways were obtained from the study of Possemato et al [32]. All of the above results were visualized using heat maps.

### 2.5. Prediction of Immunotherapy and Chemotherapy Treatment

To evaluate individual responses to immunotherapy, we used 795 specific gene sets found by other researchers in the melanoma cohort with anti-CTLA-4 or anti-PD-1 immunosuppressive therapy as a reference [33]. Subclass mapping (SubMap) was used to analyze the similarity between the risk group and the immunotherapy subgroup and identify patients who responded better to both immunotherapy agents [34]. The susceptibility to chemotherapeutic drugs was determined by estimating the half-maximum inhibitory concentration (IC_50_) of the samples using the Cancer Drug Sensitivity Genomics (GDSC) database and ridge regression analysis.

### 2.6. Statistical Analysis

The Kruskal–Wallis test was used to compare continuous data between the three groups. The relationship between these two factors was evaluated using Pearson’s correlation coefficient. The distribution of categorical variables among the groups was compared using the chi-square test. In the external validation cohort, the top 200 specific marker genes in the TCGA-ACC cohort were selected by nearest template prediction (NTP) analysis [35]. The log-rank test and K-M analysis were used to compare the survival rates of the high- and low-risk groups. Cox models were used to calculate the HR and 95% CI. The risk score relied on multivariate COX regression analysis to identify whether it had an independent prognostic effect and was bounded by *p* < 0.05. All analyses were performed using the R software (version 4.1.2) (http://www.r-project.org, Bell Laboratories, Windsor, WI, USA).

## 3. Results

### 3.1. Establishment of Molecular Subtypes

As shown in Figure 1a, when the number of clusters was three, the scores of the gap -statistical and CPI analyses were the highest. We then applied 10 multi-omics ensemble clustering algorithms to the three preset clusters and combined the results. Favorable consistency was observed among the three clusters using the ten algorithms (Figure 1b). Then, we evaluated the cluster quality via silhouette analysis, and the high silhouette width represented the robustness of the three clusters (0.74 vs. 0.64 vs. 0.47) (Figure 1c). Therefore, we redefined ACCs into three subtypes: ACC1, ACC2, and ACC3. Based on the multi-omics data in the TCGA-ACC cohort, we visualized diverse molecular features among the three subtypes, and the top ten items for each omics are listed in Figure 1d. In addition, we observed significantly different clinical outcomes among the three subtypes. ACC2 patients showed more advanced stages (59.1% vs. 15.6% vs. 50.0%, *p* = 0.022, Appendix A) and shorter overall survival, disease-specific survival, disease-free interval, and progression-free interval than ACC1 and ACC3 patients (all *p* < 0.001, Figure 2). ACC2 represented the poorest phenotype, ACC1 represented the best, and ACC3 was moderate.

### 3.2. Signaling Pathway Activation in ACC Subtypes

In 50 HALLMARK terms for each patient, we found that ACC1 patients had more immune activation, such as interferon alpha response [36], interferon gamma response [37] and Kras signal up [38]. ACC2 patients had more cell-cycle-related pathway activation, such as G2M checkpoint [39], E2F targets, and DNA repair (Figure 3a). After comparing 100 pathways related to metabolism of ACCs, it was observed that ACC1 patients were enriched in drug metabolism by other enzymes [40], retinol metabolism, and pentose and glucuronate interconversion pathways. ACC2 patients were enriched in the homocysteine cycle, the methionine cycle, and pyrimidine biosynthesis. Steroid biosynthesis [41], cholesterol biosynthesis [42], and terpenoid backbone biosynthesis pathways were enriched in ACC3 patients (Figure 3b). Specifically, pathways related to DNA repair [43] were enriched in ACC2 and ACC3 patients (Figure 3c).

According to the TMB data, we selected six genes (DST, FAT4, KMT2B, APOB, OBSCN, and ZCCHC6) with the highest mutation rates for demonstration. ACC2 patients had the highest rates of tumor mutation (Figure 4a). We then compared the base mutations of all patients; ACC2 also had the most mutations. C > T and T > C mutations were the most common (Figure 4b). However, ACCS had no obvious difference in genome copy numbers (Figure 4c), and the patients were divided into altered and unaltered groups according to the mutation data to compare their survival outcomes. The results showed that the mutation group had worse OS (Figure 4d). In addition, protein–protein interaction (PPI) enrichment analysis was performed using Metascape online tools based on the BioGrid, InWeb_IM, and OmniPath databases. The molecular complex detection (MCODE) algorithm was used to identify densely connected network components, and each MCODE component was independently enriched in different pathways and biological processes. Finally, seven pathways with the best scores were retained to describe the function of DEGs in ACCs, including cell-cycle function, mitotic nuclear division, and the PID PLK1 pathway (Figure 4e). Finally, we also found that these mutated genes were related to transcription factors, such as E2F1, TP53, E2F4, and YBX1 (Figure 4f).

### 3.3. ACC1 Patients May Benefit More from Anti-PD-1 Therapy, and Chemotherapy Is More Suitable for ACC2 Patients

To further evaluate the immune status of the three subtypes, we compared their estimated immune scores, immune cell subsets, and immune signaling molecules, which have been reported to serve as biomarkers for immunotherapy [31,44]. As shown in Figure 5a, the ACC1 subtype exhibited the highest estimated immune score (*p* = 11.1 × 10^−8^), immune cell subsets (*p* = 6.9 × 10^−8^), and immune signaling molecules (*p* = 1.6 × 10^−6^), which was consistent with the results of the pathway analysis. We investigated the infiltration landscape of different immune cells among the three subtypes. Compared to ACC2 and ACC3 subtypes, the ACC1 subtype had higher filtration of immunocytes such as central-memory CD4 T cells, plasmacytoid dendritic cells, mast cells, macrophages, regulatory T cells, activated CD8 T cells, and T helper cells. Based on 18 immune-related signatures that have been published [45], the ACC1 subtype exhibited higher immune scores in most signatures, including cytotoxic cells, T.NK. meta, CYT, treg cells, T cells, 13 T-cell signatures, TLS, WNTTGFB signatures, B cell cluster, 6 gene IFN signatures, macrophages, and MDSC (Figure 5b). PD-L1 is a special protein in tumor cells that can bind to PD1 on effector T cells to induce T cell exhaustion, which is a pivotal factor implicated in tumor immune escape [46]. We found that the ACC1 subtype expressed more PD-L1 and PD-1 than the ACC2 and ACC3 subtypes (all *p* < 0.05, Figure 5c). Therefore, the ACC1 subtype represented the high immune-infiltration phenotype, which was also called “hot tumor” and indicated a potential response to anti-PD-1 therapy [47]. Furthermore, SubMap analysis showed that patients with the ACC1 subtype would benefit more from anti-PD-1 therapy (Bonferroni *p* < 0.05, Figure 5d). In addition, we evaluated the susceptibility of the three subtypes to chemotherapy. The results showed that ACC2 patients had better sensitivity to cisplatin (*p* = 3.9 × 10^−6^), doxorubicin (*p* = 3.4 × 10^−6^), gemcitabine (*p* = 1.7 × 10^−7^), and etoposide (*p* = 3.3 × 10^−6^) (Figure 6). Collectively, this novel ACC molecular classification may facilitate the selection of appropriate treatments for different patients. The treatment of ACC1 patients with anti-PD-1 therapy and ACC2 patients with cisplatin, doxorubicin, gemcitabine, or etoposide appears to confer more clinical benefits.

### 3.4. Extra Validation for Molecular Subtypes in GEO Cohorts

To further validate the results in the TCGA-ACC cohort, three GEO cohorts were enrolled: GSE19750, GSE33371, and GSE49278. A combat algorithm was first conducted to eliminate the batch effect of the three GEO cohorts to make the data more comparable (Figure 7a). To distinguish the three subtypes, the top 300 specific genes for each subtype were selected to represent the separation of the three subtypes (Figure 7b). Consistently, we compared the OS of different subtypes and found that ACC2 patients had the worst prognosis (*p* < 0.001, Figure 7c), which was consistent with the results in the TCGA-PRAD cohort. ACC2 patients had worse survival than ACC1 and ACC3 (17.6% vs. 62.5% vs. 43.5%, *p* < 0.01) and a worse proportion of advanced-stage disease (55.9% vs. 18.7% vs. 26.0%, *p* = 0.031, Appendix A). In the pathway enrichment analysis, ACC1 patients had more activation of immune and drug-metabolism pathways, while ACC2 patients had more cell-cycle-related pathway activation, and pathways related to DNA repair were enriched in ACC2 and ACC3 patients. These functional results were similar to those of the TCGA-ACC cohort (Figure 8). Among the 18 immune-related signatures, ACC1 patients also scored highest for most items, including cytotoxic cells, T.NK. meta, CYT, treg cells, T cells, 13 T-cell signatures, TLS, B cell cluster, 6 gene IFN signatures, macrophages, and MDSC. The results showed that ACC1 represented a high immune activation phenotype and was potentially susceptible to targeted immunotherapy. Furthermore, it is worth mentioning that the ACC3 subtype also exhibited relatively higher immune scores, despite being well below ACC1 (Figure 9a). SubMap analysis showed that the patients with ACC1 were sensitive to anti-PD-1 therapy. We observed that ACC3 patients were sensitive to anti-CTLA-4 therapy (Figure 9b). Moreover, drug-sensitivity tests showed that ACC2 patients had the highest drug sensitivity to cisplatin, doxorubicin, gemcitabine, etoposide, and paclitaxel (Figure 9c) (all *p* < 0.01). In addition, our system remained an independent prognostic factor in the four ACC patient cohorts after adjustment for other major clinical characteristics (all *p* < 0.05, Table 1). This shows the reliability of our classification.

## 4. Discussion

The common age of onset for patients with ACC is between 50 and 70 years. Although most ACC is considered sporadic and the cause is unknown, a small number of cases are thought to be associated with genetic predisposition, including Lynch syndrome, Li–Fraumeni syndrome, multiple endocrine neoplasia type 1, and familial adenomatous polyposis [48,49,50,51]. Ripley et al. [52] found that the first-line treatment for recurrent or metastatic ACC depends on the patient’s underlying state and tumor characteristics. For patients who tolerate systemic chemotherapy, etoposide, doxorubicin, and cisplatin combined with mitotane (EDP-M) is superior to streptomycin–mitotane. ACC is an aggressive form of cancer, with an overall 5-year survival rate of 16–47%. The 5-year survival rates from stages I to IV were 81%, 61%, 50%, and 13%, respectively [4,53,54]. Through targeted gene analysis, mutations in TP53 or CTNNB1 have been found to be associated with molecular alterations in major ACC signaling pathways, higher tumor stage, and poorer disease-free survival (DFS). Activation of the Wnt/CTNNB1 pathway is associated with a high mitotic rate and a low survival rate. However, these markers did not show independent prognostic value in multivariate analyses, including tumor grade [55].

The immune system plays an important role in the surveillance and elimination of cancer cells, and immune evasion through various mechanisms is considered one of the characteristics of cancer [56]. Priming and activation of peripheral immune cells lead to a T-cell inflammatory phenotype, including expansion of CD8+ cytotoxic T cells, interferon signaling, and local production of chemokines [57]. Rooney et al. [28] demonstrated that cytolytic immune activity, measured by industrial expression of perforin 1 and granzyme B genes, was associated with higher mutation counts. Their prediction of antigenic epitopes in a range of solid tumor malignancies supports the idea that tumor types with a high mutational burden are more susceptible to immunotherapy strategies. Several researchers have demonstrated that tumor-infiltrating lymphocytes (TILs) are associated with improved clinical outcomes in ovarian-cancer patients [58,59,60]. Importantly, blocking PD-1, LAG-3, or CTLA-4 with gene ablation or blocking antibodies alone leads to compensatory upregulation of other checkpoint pathways, enhancing their ability to locally suppress T cells, which in turn can be overcome by a combination of blocking strategies [61].

According to Jouinot et al., the Ki67 index and targeted methylation measures of MS-MLPA can be utilized in conjunction with ENSAT staging and clinically common prognostic indicators [62]. Nevertheless, CpG islands are infrequently methylated, especially those connected to gene promoters. Further research is required to ascertain the extent to which DNA methylation of CpG islands controls gene expression [63]. The genetic changes identified in the targetable pathway indicate a potential route for novel treatments aimed at common chemotherapy-resistant cancers [64]. Dysregulation of miRNA subsets in ACC may contribute to the development of this malignancy. Additionally, it has been demonstrated that ACC patients with high expression of miRNA-related subsets have poorer survival rates, indicating the potential prognostic utility of these subsets [65]. According to certain researchers, the ACC-related genes TP53 (8 of 41 tumors, 19.5%) and CTNNB1 (4 of 41 tumors, 9.8%) both exhibited somatic mutations. Somatic mutations in recurrent ZNRF3 and TERT sites and genes created by ACC are mutually exclusive. Additionally, according to gene ontology, Wnt signaling is the most often altered pathway in ACC [66].

Researchers have discovered that lncRNA SNHG3 is associated with miR-577/SMURF1 in prostate cancer and miR-139-5p/TOP2A in renal cell carcinoma. It is also associated with CDK6, Bax, Bcl-2, N-cadherin, E-cadherin, and vimentin [67,68]. When Xp21 is lost, NR0B1, which causes X-linked AHC, GK, which causes glycerol kinase deficiency, and in certain circumstances, DMD are also lost (resulting in Duchenne muscular dystrophy). When the Xp21 deletion expanded proximally to encompass DMD or when a larger loss extended distally to include IL1RAPL1 and DMD, developmental abnormalities were observed in men with Xp21 deletion [69]. DNA methylation collaborates with histone changes and miRNAs to control transcription. Additionally, research has shown that DNA methylation controls miRNA expression [63]. Some scientists have discovered that there is no connection between transcription expression and its target factors in E. coli. Furthermore, the static gene regulatory networks (GRNs) currently in use are insufficient to explain transcriptional regulation. This suggests that, when examining the cell at a systemic level, one cannot expect to observe a causal link between the expression of transcription factors and their targets [70]. The expression of hundreds of transcripts is often cataloged by RNA-seq measurements, but most are redundant (i.e., strongly correlated) or noisy. In addition, the number of samples available is less than the number of features owing to the expenses associated with conducting experiments, which makes it simple for conventional machine learning and statistical algorithms to overfit the biological data [71].

The biological processes of a tumor are extremely complex, and different types of features are associated with each other. Therefore, it is crucial to interpret the heterogeneity of tumors using multi-omics analysis. We used 10 algorithms to determine the ACC multi-omics system by consensus clustering, which made the system more stable and convincing. We found that immune-related pathways were more activated, and drug metabolism pathways were enriched in ACC1 subtype patients. In addition, ACC1 patients are sensitive to PD-1 immunotherapy and have the lowest sensitivity to chemotherapeutic drugs. Patients with the ACC2 subtype had the worst survival prognosis and the highest tumor-mutation rate. Meanwhile, cell-cycle-related pathways, amino acid synthesis pathways, and immunosuppressive cells were enriched in ACC2 patients, steroid and cholesterol biosynthetic pathways were enriched in patients with the ACC3 subtype, and DNA-repair-related pathways were enriched in subtypes ACC2 and ACC3. We assumed that between ACC1 and ACC2, ACC3 is the transition type. According to the results, even though ACC2 and ACC3 have a comparable distribution of tumor stage, patients who belonged to ACC2 had much worse prognoses than those who belonged to ACC3. In comparison to other subtypes, ACC3 exhibits more WNT pathway activation, greater steroid and cholesterol production, greater copy number change, and a lack of the OBSCN and ZCCHC6 mutations that were found in ACC1 and ACC2. The lowest level of immune pathway activation is then met by the ACC3 subtype. The sensitivity of the ACC2 subtype to cisplatin, doxorubicin, gemcitabine, and etoposide was better than that of the other two subtypes. For 5-fluorouracil, there was no significant difference in the sensitivity to paclitaxel between the three groups. We believe that multi-omics analysis in ACC can provide patients with more accurate clinical treatment and better prognosis prediction.

## Figures and Tables

**Figure 1 cells-11-03784-f001:**
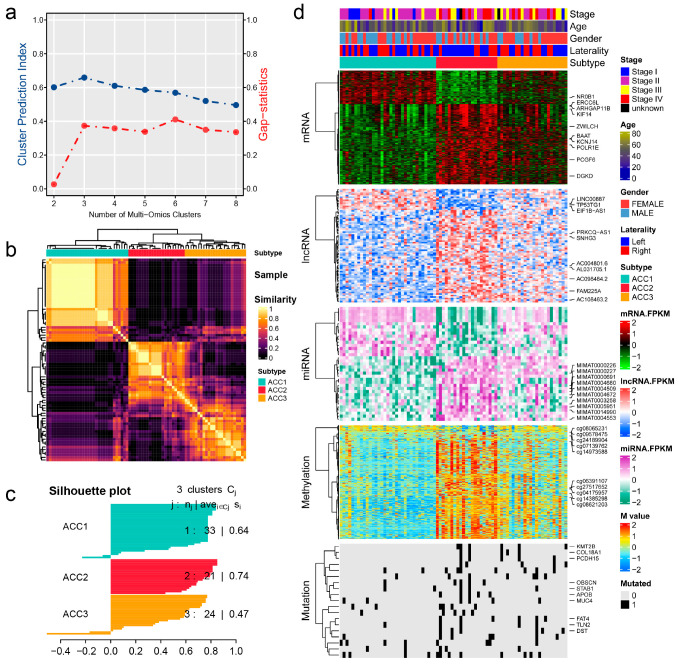
Recognition of the adrenocortical carcinoma multi-omics classification system in the TCGA-ACC cohort. (**a**) CPI analysis and gap-statistical analysis results. (**b**) Consensus matrix for three clusters based on the 10 algorithms. (**c**) Silhouette-analysis evaluation of cluster quality. (**d**) Visualization of multi-omics data for mRNAs, lncRNAs, miRNAs, DNA CpG methylation sites, and mutant genes.

**Figure 2 cells-11-03784-f002:**
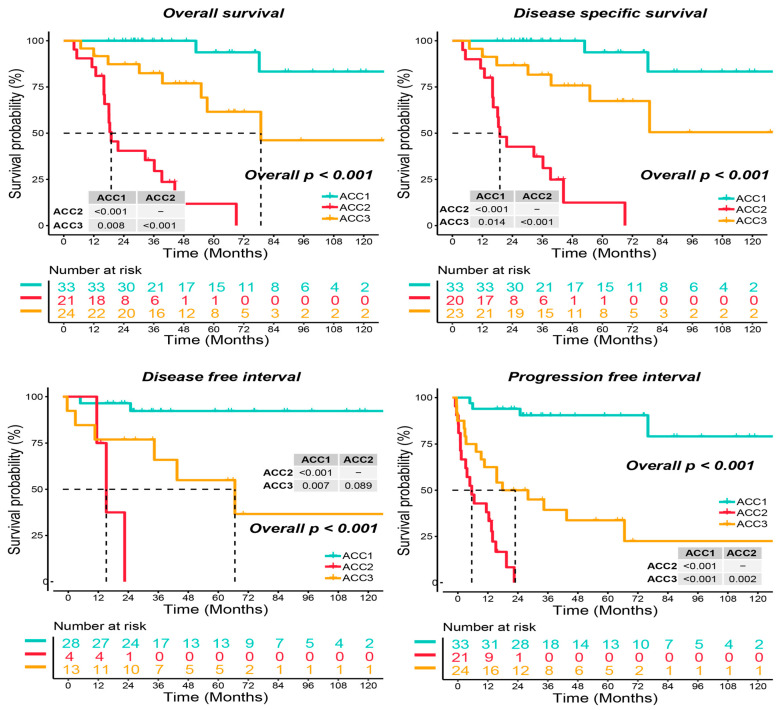
Differential survival outcome in three ACC subtypes, log-rank test.

**Figure 3 cells-11-03784-f003:**
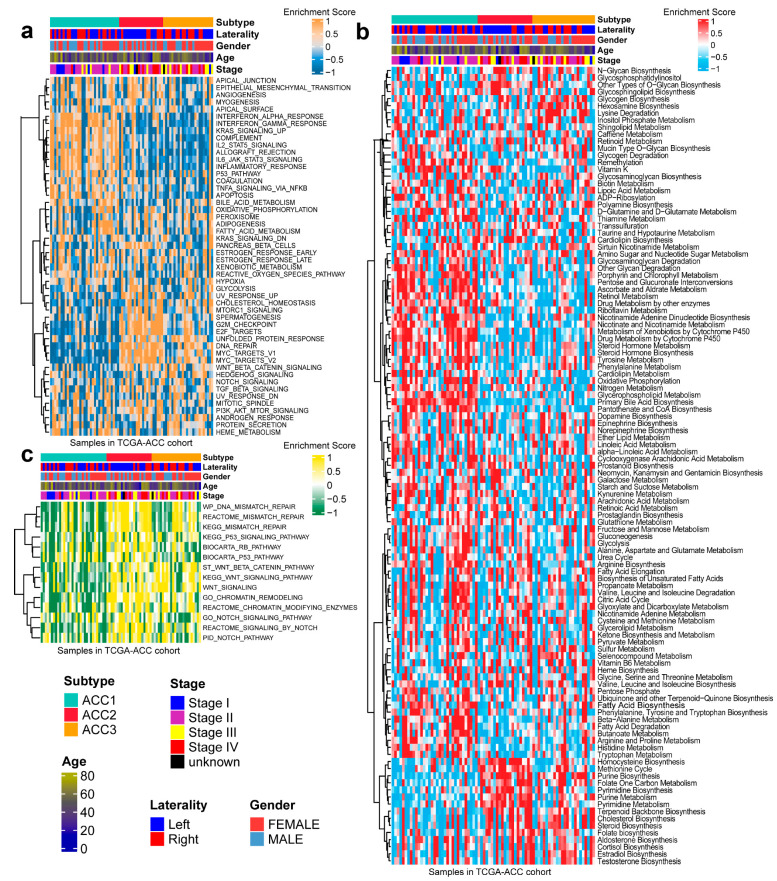
Differential activity of tumor-associated pathways across three ACCs subtypes in TCGA-ACC cohort. (**a**) Heatmap of 50 differentially activated HALLMARK pathways. (**b**) Heatmap of 100 pathways related to metabolism of ACCs. (**c**) Heatmap of DNA repair pathways.

**Figure 4 cells-11-03784-f004:**
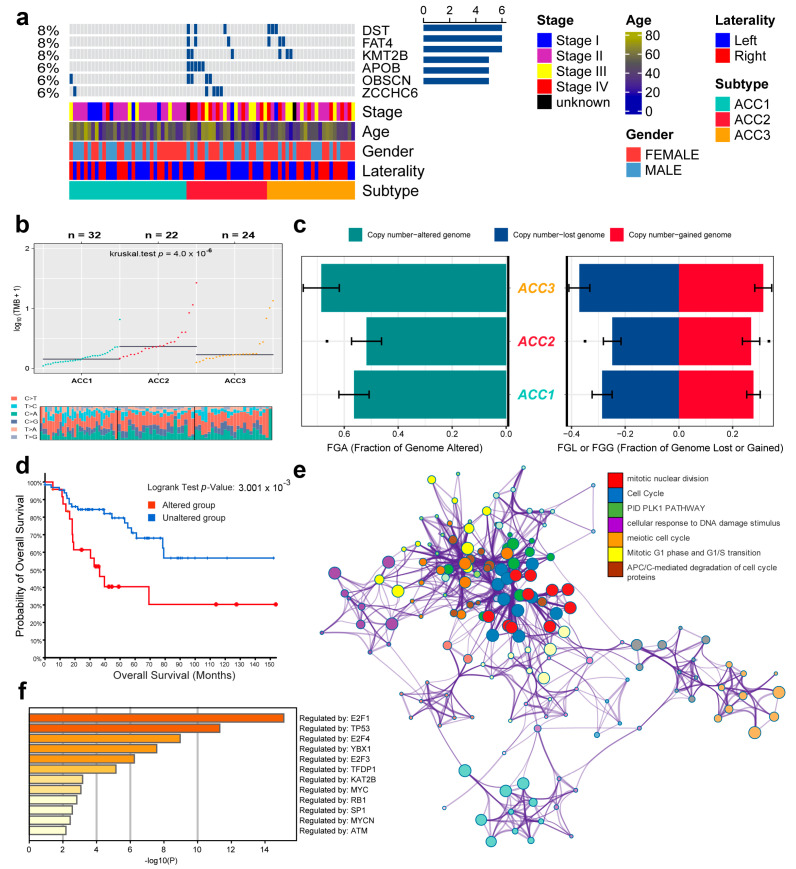
Differential tumor-mutation burden and copy number in three ACC subtypes. (**a**) Mutations in the six genes with the highest mutation rates. (**b**) Base mutations in the three ACCs cohorts. (**c**) Differences in genome copy numbers. (**d**) Differences in overall survival between mutant and non-mutant groups. (**e**) Functional enrichment analysis of the six genes with the highest mutation rate. (**f**) Relationship between mutation genes and transcription factors.

**Figure 5 cells-11-03784-f005:**
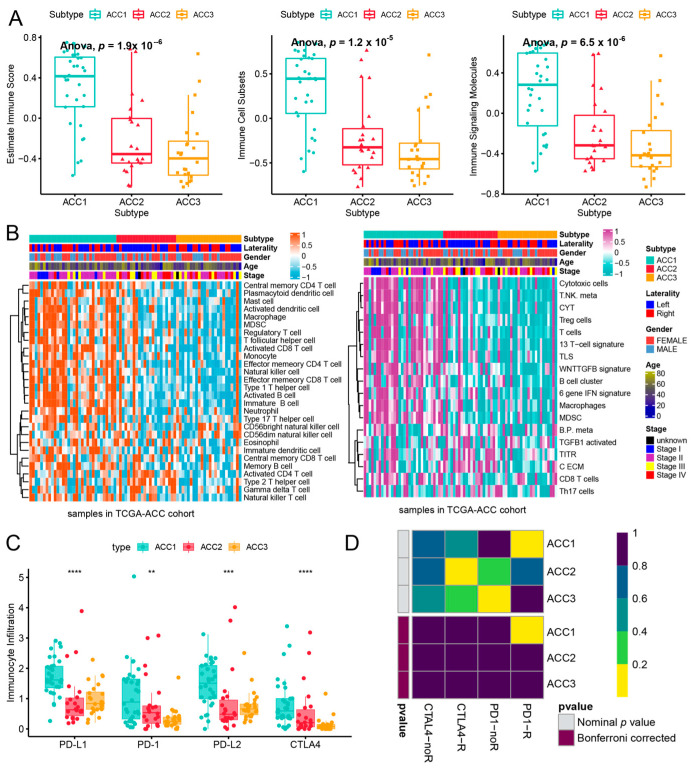
Differences of immune infiltration in three ACC subtypes. (**a**) Score of related indicators of immune infiltration. (**b**) Heatmap of infiltration landscape of different immune cells between the three subtypes. (**c**) Expression of the three subtypes to PD-L1 and PD-1. (**d**) SubMap analysis of anti-PD-L1 therapy. ** *p* < 0.01, *** *p* < 0.001, **** *p* < 0.0001.

**Figure 6 cells-11-03784-f006:**
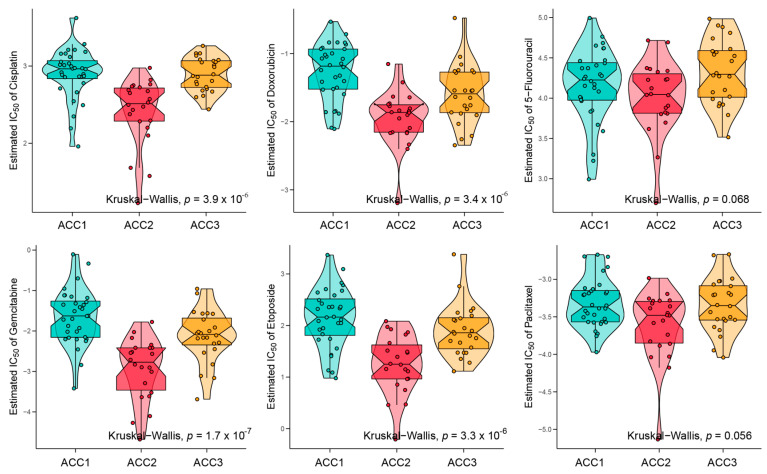
Differences in chemotherapy susceptibility between the three subtypes.

**Figure 7 cells-11-03784-f007:**
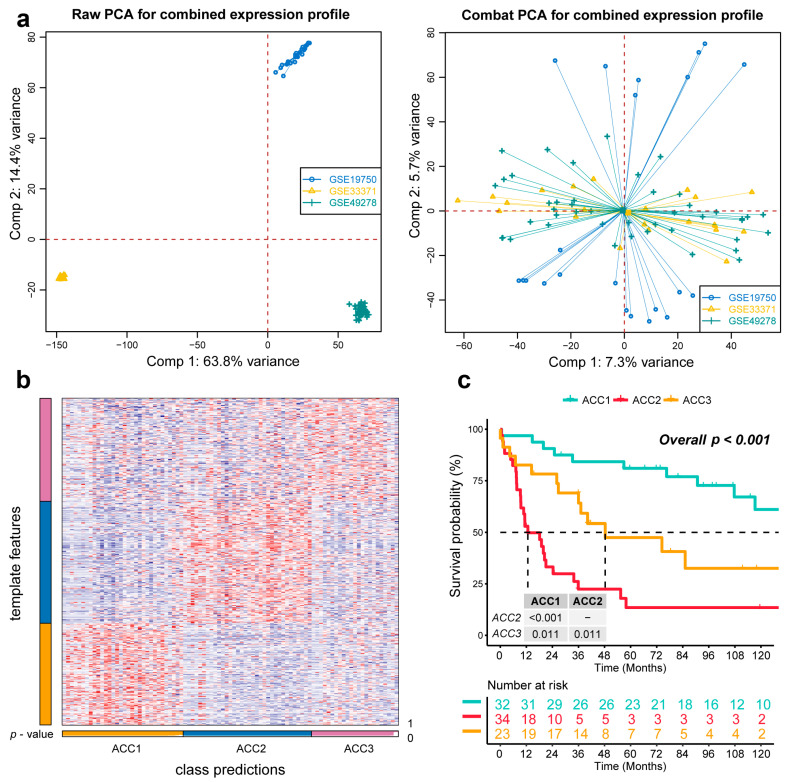
Recognition of the adrenocortical carcinoma multi-omics classification system in the GEO cohort. (**a**) The combat algorithm eliminates the queue batch effect. (**b**) Representing the ACCs in the GEO cohort. (**c**) Differential overall survival outcome in reproduced ACCs of GEO cohorts, log-rank test.

**Figure 8 cells-11-03784-f008:**
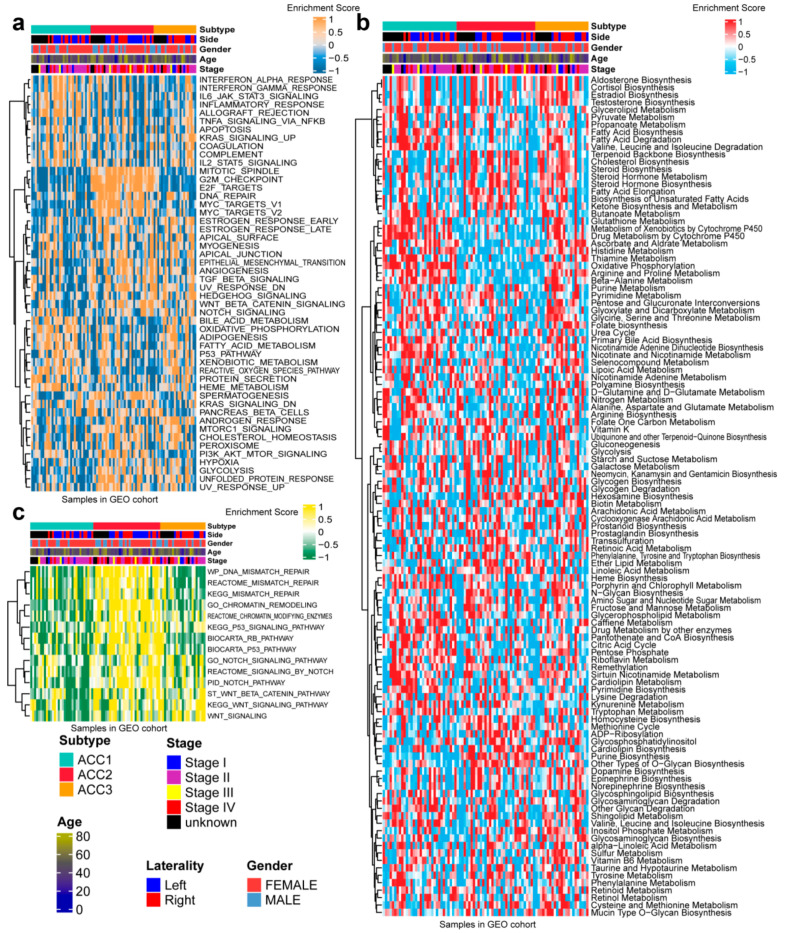
Differential activity of tumor-associated pathways across three ACCs subtypes in GEO cohort. (**a**) Heatmap of 50 differentially activated HALLMARK pathways. (**b**) Heatmap of 100 pathways related to metabolism of ACCs. (**c**) Heatmap of DNA repair pathways.

**Figure 9 cells-11-03784-f009:**
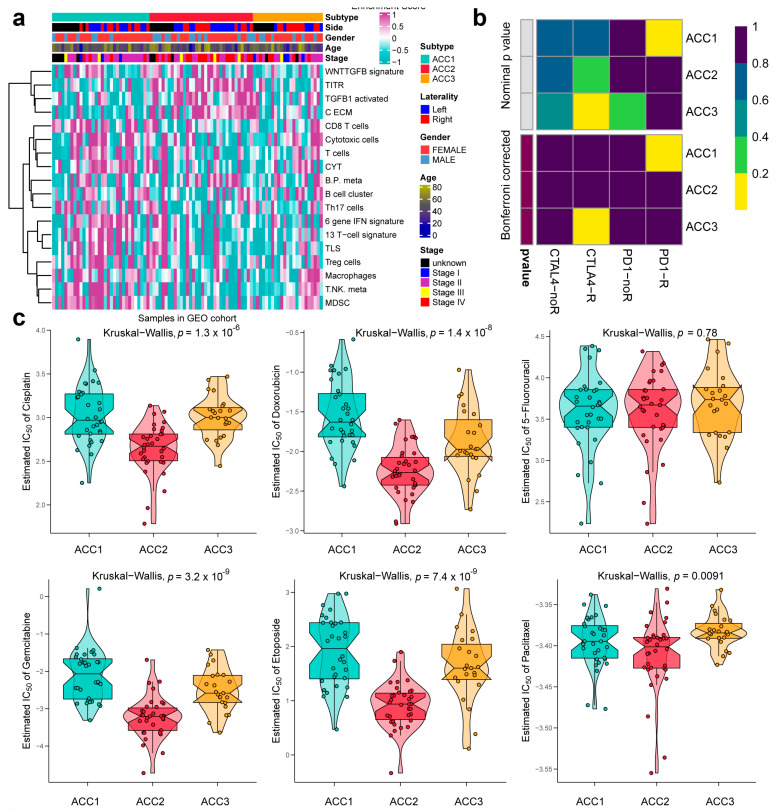
Differences in immune infiltration and chemotherapy susceptibility of three subtypes in GEO cohort. (**a**) Heatmap of infiltration landscape of different immune cells between the three subtypes. (**b**) SubMap analysis of anti-PD-L1 therapy. (**c**) Differences in chemotherapy susceptibility between the three subtypes.

**Table 1 cells-11-03784-t001:** Prognostic value of ACC subtype after adjusting for clinicopathological parameters.

	HR	95% CI	*p* Value
TCGA-ACC Cohort			
Age	1.013	(0.986–1.041)	0.351
Gender, male vs. female	1.455	(0.614–3.451)	0.394
Laterality, right vs. left	1.533	(0.664–3.54)	0.317
Stage			
Stage II vs. stage I	2.883	(0.301–27.628)	0.358
Stage III vs. stage I	6.28	(0.641–61.55)	0.115
Stage IV vs. stage I	16.164	(1.49–175.301)	0.022
Stage unknow vs. stage I	2.451	(0.117–51.349)	0.564
ACC subtype			
ACC2 vs. ACC1	45.146	(7.393–275.694)	0
ACC3 vs. ACC1	4.661	(0.877–24.779)	0.071
GEO cohort			
Age	1.01	(0.99–1.031)	0.31
Gender, male vs. female	1.236	(0.649–2.354)	0.518
Laterality			
Right vs. left	1.15	(0.54–2.45)	0.717
Unknow vs. left	1.2	(0.51–2.822)	0.677
Stage			
Stage II vs. stage I	2.765	(0.351–21.81)	0.334
Stage III vs. stage I	8.223	(0.909–74.351)	0.061
Stage IV vs. stage I	11.723	(1.504–91.399)	0.019
Stage unknow vs. stage I	4.067	(0.453–36.548)	0.21
ACC subtype			
ACC2 vs. ACC1	4.959	(2.241–10.97)	0
ACC3 vs. ACC1	2.578	(1.048–6.341)	0.039

## Data Availability

Multi-omics data of ACC patients including DNA methylation, gene mutations, mRNA, miRNA, and LncRNA were downloaded from TCGA-ACC dataset (https://portal.gdc.cancer.gov, accessed on 15 April 2022) and GEO dataset (https://www.ncbi.nlm.nih.gov/geo/, accessed on 15 April 2022); the miRNA expression and DNA methylation 450 matrix were download from the UCSC Xena (https://xenabrowser.net/datapages, accessed on 15 April 2022). Download the somatic mutation data from cbiopportal (https://www.cbioportal.org/, accessed on 15 April 2022). Further enquiries can be directed to the corresponding author. We declare that the data and materials in this study will be provided free of charge to scientists for noncommercial purposes.

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
