# Peer review of "Molecular Cluster Mining of Adrenocortical Carcinoma via Multi-Omics Data Analysis Aids Precise Clinical Therapy"

_cells, 2022, doi:10.3390/cells11233784_

Round 1
Reviewer 1 Report
1. Major concerns
a. Introduction
i. The introduction didn’t provide enough background on multi-omics to study diseases, especially cancer topics and ACCs. Why is it essential to use multi-omics? Why using one modality, e.g., gene expression, is not sufficient?
b. Data
i. It is unclear what the distribution is in terms of modalities of the datasets used in discovery (TCGA) and validation (GEO). A specific question is: Does every sample have all five modalities in both datasets? This point has to be clearly stated or justified if the answer to the first question is a ‘no.’
ii. It looks like GEO data has only the expression data, one modality. The authors used only gene expression, 300 specific genes, to distinguish the subtypes. If this is indeed accurate, it means that there is no need to use the other four modalities;
c. Methods
i. Many methods mentioned in the paper lack brief introductions of:
1. Why does it work?
2. What is the input?
3. What is the output?
4. How to interpret the result?
5. References
Here are some of the methods that lack some or all of the above information: MOVICS (line 89), Gaps statistics, and Cluster prediction index (CPI) (line 93).
ii. It is not clear how the ten clustering methods are combined using a consensus set (line 94), as Figure 2B is also not clear. It lacks essential text legend to explain the figure, a common problem for all other figures.
iii. It does not make sense to calculate a correlation between the t.test result and the Wilcoxon test (line131). The criteria for selecting which statistical test to use should depend on the data distribution. It would help if you used a Q-Q plot to decide the normality of your data distribution and which test to use.
iv. Many procedures mentioned in the results are not clearly stated in the Methods section. For example, SubMap and immune infiltration estimation.
d. Results
i. It looks like ACC3 does not form a cluster individually. It is more like a mixture of ACC1 and ACC2, according to the figures (Fig. 1D, Figure 3, Fig. 4B-C, etc.)
ii. Despite the ACC3, one significant difference in the ACC1 and ACC2 is the composition of ACC stages, which was significant in Table 1 (p=0.02) and Figure 1D. ACC stage is a very important predictor of the overall survival rate. Considering ACC3 might be a mixture of ACC1 and ACC2, the subtypes the authors found might be highly confounded by the ACC stages. Thus, the novelty of this paper is doubted.
iii. All the figures’ legends lack essential illustrations to understand. The x-axis, y-axis, colors, values, etc. are, needed to be clarified in the legends.
iv. Results and discussion section lack discussion on each of the modalities. For example, CpG data is rarely mentioned or discussed why it is not essential.
2. Minor concerns:
a. It is unclear what ‘HALLMARK terms’ are. Please define it or provide a reference (line 109).
b. Figure 5C: the legend is ‘The response of the three subtypes to PDL1 and PD1’. However, the text reads that ACC1 has a higher expression of PDL1 and PD1. Is ‘response’ the same as ‘higher expression’?
c. There are some typos and fragments of sentences in the paper. The authors need to revise those typos.
d. Some places lack references.
i. ssGSEA needs a reference (line 109).
ii. Line 216-217: “which have been published”
Author Response
Dear Reviewers and editors,
First of all, thank you very much for reviewing my manuscript and putting forward valuable modification suggestions. We accepted your opinions seriously and made an extensive revision. Therefore, the process is complicated, but at least now it is all completed. We feel very worthwhile, and thank you for your valuable advice again.
We have resubmitted a manuscript version. The major corrections in the paper and the responses to your comments are as follows:
Please see the attachment.

Reviewer 2 Report
In this manuscript, the authors (Guan et al,) analyzed the multi-omics data of ACC, including clinical and pathological features, genomic mutations, DNA methylation, mRNA, LncRNA, miRNA, and proteomic data of ACC patients. They found several interesting features correlated to different stages of ACC, including activated immune-related pathways and enriched drug metabolism pathways in ACC1 patients, enriched cell cycle-related pathways, amino acid synthesis pathways and immunosuppressive cells in ACC2 patients and enriched steroid and cholesterol biosynthetic pathways in ACC3 patients. This study revealed several oncogenic pathways that haven't been reported to relate to ACC. These observations offered new clues for the prognosis and treatment of ACC.
I have several minor suggestions and questions here:
1, In the discussion part, could the author offer more detail about the inconsistency of different omics data? How to interpret them? Is there any priority for these data?
2, In the result part, could the author show some examples of enriched pathways in ACC patients which have already been reported in other papers?
3, It seems that the tables are not very informative in the Main Text. Could the authors move them to the supporting information?
Author Response
Dear Reviewers and editors,
First of all, thank you very much for reviewing my manuscript and putting forward valuable modification suggestions. We accepted your opinions seriously and made an extensive revision. Therefore, the process is complicated, but at least now it is all completed. We feel very worthwhile, and thank you for your valuable advice again.
We have resubmitted a manuscript version. The major corrections in the paper and the responses to your comments are as follows:
Please see the attachment

Round 2
Reviewer 1 Report
The authors have addressed most of my comments. However, I still have some minor concerns that could improve the quality of this paper.
1. The discussion on ACC-3 is lacking in the discussion section. Since the primary finding of this study is the 3 ACC subtypes, it is essential to clearly state the differences between the three subtypes and the potential transitional relationships between one and the others.
In the "response to reviewers' comments" file, the author wrote the following sentences: (1) We can regard that ACC3 is the transition type between ACC1 and ACC2. (2) Therefore, ACC3 is significantly different to ACC2 and ACC1.
In the discussion section, lines 383-393, the authors discussed a lot on ACC1 and ACC2, which is good because they have more diagnostic and clinically relevant features according to the results. However, the major differences discussed on the ACC3 are not sufficient. The authors thought ACC3 could be regarded as a transition type, which might also be an essential and novel finding but not discussed in the paper.
2. Line 325, DFS, is not defined previously. It would help if the authors could replace it with the unabbreviated term.
Author Response
Response to Reviewer 1 Comments
Dear Reviewers and editors,
First of all, thank you very much for reviewing my manuscript and putting forward valuable modification suggestions. We accepted your opinions seriously and made an extensive revision. Therefore, the process is complicated, but at least now it is all completed. We feel very worthwhile, and thank you for your valuable advice again.
We have resubmitted a manuscript version. The major corrections in the paper and the responses to your comments are as follows:
The authors have addressed most of my comments. However, I still have some minor concerns that could improve the quality of this paper.
- The discussion on ACC-3 is lacking in the discussion section. Since the primary finding of this study is the 3 ACC subtypes, it is essential to clearly state the differences between the three subtypes and the potential transitional relationships between one and the others.
In the "response to reviewers' comments" file, the author wrote the following sentences: (1) We can regard that ACC3 is the transition type between ACC1 and ACC2. (2) Therefore, ACC3 is significantly different to ACC2 and ACC1.
In the discussion section, lines 383-393, the authors discussed a lot on ACC1 and ACC2, which is good because they have more diagnostic and clinically relevant features according to the results. However, the major differences discussed on the ACC3 are not sufficient. The authors thought ACC3 could be regarded as a transition type, which might also be an essential and novel finding but not discussed in the paper.
Response:
Dear reviewer,
Thanks for your careful review, we agree with your suggestion and had added the related words in discussion section. The details are as follows:
The biological processes of a tumor are extremely complex, and different types of features are associated with each other. Therefore, it is crucial to interpret the heterogeneity of tumors using multi-omics analysis. We used 10 algorithms to determine the ACC multi-omics system by consensus clustering, which made the system more stable and convincing. We found that immune-related pathways were more activated, and drug metabolism pathways were enriched in ACC1 subtype patients. In addition, ACC1 patients are sensitive to PD-1 immunotherapy and have the lowest sensitivity to chemotherapeutic drugs. Patients with the ACC2 subtype had the worst survival prognosis and the highest tumor mutation rate. Meanwhile, cell cycle-related pathways, amino acid synthesis pathways, and immunosuppressive cells were enriched in ACC2 patients. Steroid and cholesterol biosynthetic pathways were enriched in patients with the ACC3 subtype. DNA repair-related pathways were enriched in subtypes ACC2 and ACC3. We assumed that between ACC1 and ACC2, ACC3 is the transition type. According to the results, even though ACC2 and ACC3 have a comparable distribution of tumor stage, patients who belonged to ACC2 had much worse prognoses than those who belonged to ACC3. In comparison to other subtypes, ACC3 exhibits more WNT pathway activation, greater steroid and cholesterol production, greater copy number change, and a lack of the OBSCN and ZCCHC6 mutations that were found in ACC1 and ACC2. The lowest level of immune pathway activation is then met by the ACC3 subtype. The sensitivity of the ACC2 subtype to cisplatin, doxorubicin, gemcitabine, and etoposide was better than that of the other two subtypes. For 5-fluorouracil, there was no significant difference in the sensitivity to paclitaxel between the three groups. We believe that multi-omics analysis in ACC can provide patients with more accurate clinical treatment and better prognosis prediction.
- Line 325, DFS, is not defined previously. It would help if the authors could replace it with the unabbreviated term.
Response:
Dear reviewer,
Thanks a lot for your advice. We had defined the “DFS” for “disease free survival” in the main text.
